# Thyroid Disorders Spectrum in Pediatric Endocrine Clinic; Seven-Year Experience of a Teaching Hospital in Saudi Arabia

**DOI:** 10.3390/children10020390

**Published:** 2023-02-16

**Authors:** Mohammad H. Al-Qahtani, Sufian A. ElYahia, Abdulaziz S. AlQahtani, Abdulrahman J. AlQahtani, Abdulaziz A. Alamer, Sultan M. AlQahtani, Abdullah A. Yousef, Waleed H. Albuali, Bassam H. Awary, Ala’a A. Aldajani, Mohammed A. Al Ghamdi

**Affiliations:** 1Department of Pediatrics, King Fahd Hospital of the University, College of Medicine, Imam Abdulrahman Bin Faisal University, Dammam 34221, Saudi Arabia; 2College of Medicine, Imam Abdulrahman Bin Faisal University, Dammam 34221, Saudi Arabia

**Keywords:** thyroid disorders, children, congenital hypothyroidism, thyroiditis, subclinical hypothyroidism, hyperthyroidism

## Abstract

Thyroid disorders constitute one of the major endocrine disorders in pediatric service. It includes a range of congenital versus acquired anatomic and/or functional thyroid diseases in growing children that has a spectrum of severity from severe intellectual disability effect to subclinical mild pathologies. This study was designed to analyze the demographic characteristics, clinical pattern, and severity of thyroid disorders in the pediatric endocrine clinic patients at the teaching hospital of the university over a 7-year duration. A total number of 148 patients with thyroid disorders were seen in pediatric Endocrine clinic during the time between January 2015 and December 2021. Female patients constitute 64% of them. Acquired Hypothyroidism was the commonest disorder; 34% of the cases followed by the congenital hypothyroidism (CH), then Hashimoto’s thyroiditis, and 5.8% for others. While a very small percentage was acquired hyperthyroidism. The majority of referrals were from dermatology and other service for the screening of thyroid disease as association with other autoimmune diseases with percentage of 28.3%. Next was neck swelling manifestation in 22.6%. Thyroid disorders in children, both congenital and acquired, constitute an important medical issue for pediatricians to be aware of its variable presentations, and its potential serious health consequences on the affected children if not diagnosed and treated earlier. Acquired hypothyroidism constitutes more percentage of the thyroid disorders followed in the pediatric endocrinology outpatient clinics. Congenital hypothyroidism is the second most common thyroid disorder in the outpatient unit, having the most potential complications. These results support the international studies with the female predominance in most of thyroid disorders.

## 1. Introduction

Endocrine disorders in children are common component of the medical health issues in pediatric age group. Next to Type 1 diabetes mellitus and short stature, thyroid disorders are the most prevalent pediatric endocrine illnesses. The thyroid gland produces a thyroid hormone, which has a major role in basal metabolic process, physical growth, brain function, oxygen consumption enhancement, and neuronal and musculoskeletal development [1].

The broad spectrum of thyroid problems range in severity, and can progress to a serious level, like what happens in congenital hypothyroidism because of a failure of brain maturation that causes intellectual disability. Different genetic, syndromic, or metabolic co-morbidities may be linked to thyroid disorders [2]. Autoimmune thyroid disorders, such as Hashimoto’s thyroiditis with hypothyroidism and Graves’ disease, may present as an isolated autoimmune disease or as association with other autoimmune diseases, such as Diabetes mellites type1, celiac disease, pernicious anemia, and other autoimmune associations. [3].

Isolated or combined autoimmune thyroid disease is more prevalent in the developed countries [4], while Iodine deficiency is considered as the commonest cause of thyroid disorders worldwide [5]. Furthermore, females were found to have thyroid disorders more than males [6].

Congenital hypothyroidism (CH), which is a birth diagnosis of thyroid hormone deficiency, considered as the commonest preventable etiology of intellectual disability in human beings, with a global incidence of 1:2000 to 1:4000 newborns [1]. Majority of CH cases are classified to primary or secondary congenital thyroid gland disorders. The primary CH is a result of defects in embryological development defects of the thyroid gland, called dysgenesis, or it might be due to less common etiology duet to defect in thyroid hormone synthesis known as dyshormonogenesis. The secondary CH or central congenital hypothyroidism is caused by abnormal pituitary gland function or anatomy leading to deficient thyroid stimulating hormone (TSH), in most of the cases, it is part of the other pituitary hormones deficiency, termed as congenital hypopituitarism. Once diagnosed and confirmed CH should be started on life-long treatment with the synthetic thyroid hormone thyroxine to maintain brain and body organs appropriate growth and functions [1]. In Saudi Arabia, there was large screening for congenital hypothyroidism in neonates by Ministry of Health, which was a cord-blood test based on TSH level with cut point of 30 mIU/mL or more to be confirmed by venous blood test. They screened over 1 million newborns with overall incidence of 1:3292 with some variation in incidence in the different regions of the kingdom [7].

Down syndrome and Turner syndrome are the most common chromosomal diseases associated with thyroid disorders [8]. Down syndrome pediatric subjects specifically are more prone to have different thyroid disorders, including subclinical hypothyroidism, congenital hypothyroidism, and thyroid autoimmunity, such as Hashimoto’s thyroiditis. The issue of treating Down syndrome pediatric patients with subclinical hypothyroidism was debatable, and with different range of findings from reported mild improvements in motor development and height in treated infants in some studies to limited effect of mild increase in FT4 with no change in TSH level compared to controls [9]. There were various studies regarding epidemiological aspects of thyroid disorders in pediatric age group nationally at country level. few studies focused more on thyroid disorders, with elaborating on each single disease. In screening program done widely in the United States to detect cases of congenital hypothyroidism, it is estimated that of every 1500 to 3000 newborn, one has congenital hypothyroidism [10], and by early diagnosis, treatment is followed to ensure normal development. Screening for congenital hypothyroidism in all newborns is crucial for the early diagnosis and management of congenital hypothyroidism and it is implemented in most developed countries [10,11]. Acquired hypothyroidism is another important cause of thyroid disfunction not only in adults, but also in children and adolescents, Hashimoto’s thyroiditis is the most common cause of acquired hypothyroidism [6], with female predominance of 4.2:1 with prevalence of 1–2 % regarding autoimmune thyroiditis [12].

The major hyperthyroidism etiology in pediatric age group is Graves’ disease [13], it is less common in the pediatric age group compared to adults, its prevalence varies among different countries from 1/10,000 in the United States to 1/100,000 in the UK and Ireland. Likewise, to hypothyroidism, Graves’ disease is 4–5 times more common in females compared to males, peaking during puberty; however, it is rarely reported below the age of 4 years. Hyperthyroidism in children might present with goiter, palpitations, weight loss, headache, sleep disturbances, anxiety, and heat intolerance [14]. Its diagnosis is based on patient’s symptoms and clinical signs, laboratory confirmatory tests show decreased level of thyroid stimulating hormone TSH combined with elevated free T4 level, with or without presence of TSH-receptor antibodies. Treatment of hyperthyroidism in children is of limited options including methimazole as oral medication followed by surgery, and radioactive iodine.

Due to the scarceness of published studies in outpatient thyroid disorders in pediatric age group, the aim of this study was to analyze the pattern and variation of thyroid disorder spectrum in pediatric endocrine clinics over a 7-year duration at King Fahd Hospital of the University in Al-Khobar, Eastern Province, Saudi Arabia compared to other national and international similar studies. This might help pediatricians for better understanding of its effect on the children growth and wellbeing; in other hand, it may facilitate improving healthcare institute for the distribution and priorities of medical services setup in the outpatient departments.

## 2. Materials and Methods

### 2.1. Objectives

The objectives of our study were: (1) to determine the prevalence of thyroid disorders among pediatric patients visiting the endocrine clinic in the teaching hospital; (2) to differentiate the spectrum of different thyroid disorder in gender perspectives and among each age group of the children; and (3) to evaluate the effect of associated disorders and the major risk factors of thyroid disorder in pediatric age group.

### 2.2. Participants

The study included 148 pediatric participants, age from 1 year till 16 years. visiting the pediatric endocrinology clinic, which is a weekly clinic in the teaching hospital in the period from January 2015 to December 2021.

### 2.3. Procedure and Approval

A retrospective single-center review of all the electronic records of the patients the ICD 10 coding system and reviewing their records; therefore, the total number before applying exclusion criteria were 148 patients. Inclusion criteria includes all pediatric age group: day 1 to 16 years of age, both genders, and have thyroid disorders. Exclusion criteria were older than 17 years of age, incomplete charts, and missing data. 

The congenital hypothyroidism was defined as any patient got positive screening test for neonatal congenital hypothyroidism and confirmed by laboratory blood tests, and started on levothyroxine treatment within the 1st month of life. Hashimoto’s thyroiditis was defined by high TSH, low FT4 associated with high antithyroid peroxidase TPO and/or antithyroglobulin TG antibodies. Acquired non-Hashimoto’s hypothyroidism was defined by high TSH and low FT4 in any patient above 2 years in negative anti-TPO and/or anti TG antibodies. Hyperthyroidism was defined by low TSH and High FT4 with or without high FT3, while Graves disease was labeled if in addition had positive thyrotropin receptor antibodies, TRAb. Patients were labeled as subclinical hypothyroidism if found to have as high TSH and normal FT4 level.

The research was granted the ethical approval from the hospital institutional review board (IRB) for studies involving human participants in Imam Abdulrahman Bin Faisal University in Dammam, Saudi Arabia, IRB number IRB-UGS-2018-01-230.

### 2.4. Data Analysis

The data analysis of our study was performed utilizing the Statistical Package of Social Sciences (SPSS) version 26 (IBM Corp., Armonk, NY, USA). The results were analyzed in a descriptive statistical way to express the findings in frequencies and percentage among the different age groups and the gender differences in each type of the thyroid disorder spectrum.

## 3. Results

During the period from January 2015 to December 2021, a total of 148 included patients were following regularly at the teaching hospital of the university at eastern province. Most of the patients were Saudis 84.9%, and the rest were non-Saudis 15.1%. Majority of the patients were females 64.2% (Table 1).

Most of the patients in our study have Hashimoto’s hypothyroidism 50.34%, followed by congenital hypothyroidism 39.6%, non-Hashimoto’s hypothyroidism 25.17%, followed by subclinical hypothyroidism 15.10%; total number of cases of hyperthyroidism were 12.5%, including Graves’ disease was 10.2%, and other non-defined cases of hyperthyroidism was 26%. Thyroglossal duct cyst cases compose 6.4%, and the least number of cases in our sample was thyroid nodules, representing only 1.1% (Figure 1).

The female patients, most have Hashimoto’s thyroiditis and Congenital Hypothyroidism, and they were found to be equal in number The presentations of patients with Hashimoto’s thyroiditis were mainly hypothyroidism symptoms in half of them, goiter in one-fourth, and by screening test due to family history in one-fourth, we had only one patient who had history suggestive of hyperthyroidism and confirmed by thyroid function tests. In the other hand, most male patients were diagnosed with Congenital Hypothyroidism and non-Hashimoto’s hypothyroidism, followed by Subclinical Hypothyroidism, and none of the male patients were diagnosed with Hyperthyroidism. (Figure 2).

We categorized the patients according to the age at diagnosis and in relation to the disease into three age groups Table 2, first from birth–6 years old, which was the largest group 46.2%, followed by the second age group from 7 years old–12 years old 39.6%. The smallest was the third age group from 12 years old to 16 years old, comprising 14.2%of the total patient number. 

Congenital hypothyroidism was diagnosed in most patients between the ages of birth and 6 years old, accounting for 57.1% of cases, followed by non-hypothyroidism Hashimoto’s (24.5%), Hashimoto’s thyroiditis (8.2%), subclinical hypothyroidism (8.1%), and only one case of a thyroglossal duct cyst in this age group. Regarding the presence of antibodies, a total of 54 patients were found to have positive antibodies, which represent 36.8% of total patients.

Correlating Hashimoto’s thyroiditis to each specific antibody, we found 16 out of 18 patients diagnosed with Hashimoto’s thyroiditis 88.9% were found to have positive antithyroglobulin antibody test. A total of 13 out of 18 patients diagnosed with Hashimoto’s thyroiditis 72.2% were found to have positive thyroid peroxidase antibody test.

The second age group, which included patients aged seven to twelve years old, comprised 39.6% of the total number of patients. Acquired hypothyroidism was the most common diagnosis in this group, accounting for 45.2% of cases, followed by Hashimoto’s thyroiditis at 23.8%, acquired hyperthyroidism at 9.5%, subclinical hypothyroidism at 7.1%, and finally, subclinical hyperthyroidism and thyroglossal duct cyst at 2.4%.

The third age group who were from the age of 12 years to 16 years old (14.2%), most of them were diagnosed with acquired hypothyroidism (33.3%), followed by Hashimoto’s thyroiditis (26.7%), subclinical hypothyroidism (20%), Grave’s Disease (13.3%), and lastly, one patient was diagnosed with thyroid nodule in this age group that contribute to 6.7%.

In contrast, all the patients of congenital hypothyroidism were diagnosed at the first age group from birth to 6 years of age 100%. Most of the patients with Hashimoto’s thyroiditis were diagnosed at the second age group, 55.6% which starts from 7 years old to 12 years old patients. 

All patients with acquired hyperthyroidism including Hashimoto’s thyroiditis and non-Hashimoto’s thyroiditis were diagnosed at the second age group from 7 years old to 12 years old, 100%. Most of the patients with acquired hypothyroidism were diagnosed at the second age group from 7 years old to 12 years old 52.8%. Only one patient with subclinical hyperthyroidism was found in this second age group. 

The first age group from birth to 6 years of age and the second age group from 7 years old to 12 years old, were found to have the same number of patients diagnosed with subclinical hypothyroidism at these 2 age groups 36.4. The two patients with thyroglossal duct cyst were diagnosed at the third age group from 12 years old to 16 years of age. We got three patients with thyroid nodule were diagnosed at the second age group from 7 years to 12 years of age.

The family history of thyroid spectrum disorders performs an important role in predicting the susceptibility of the children to certain thyroid diseases, such as autoimmune thyroid disorders, or the inherited type of congenital hypothyroidism, dyshomonogenesis, which is an autosomal recessive disease. Most of the affected patients were having negative family history 66%, and those with positive family history of 34% all had positive family history of autoimmune thyroid disorders in their first or second degree relatives.

Regarding reason for referral to pediatric endocrine clinic, most patients were referred due to symptoms that suggest thyroid abnormality. Complaints were categorized into seven main categories in sequence of their percentage; the largest group is the screening tests for thyroid disorders, such as congenital hypothyroidism at birth, and the screening for autoimmune thyroid disease in the susceptible patients, followed by localized symptoms related to the thyroid gland size and shape, growth related growth parameters abnormalities involving height and weight cases, metabolic, and energy that interfere with daily activities, category of dermatological presentations, skin and hair manifestation that is related to the thyroid functions, incidental category when findings abnormal laboratory thyroid function test results without active complaint, and the last category is the gynecological symptoms related to pubertal abnormalities and interfere with female menstrual physiology (Table 3).

In relation between acquired hypothyroidism and related comorbidities, of 36 patients, 20 patients have other comorbidities. The most frequently encountered is Down Syndrome, followed by Bronchial Asthma BA, and Type 1 Diabetes Mellitus T1DM; (Table 4) out of the patients that have Hashimoto’s thyroiditis, almost two-thirds of them were having other comorbidities.

## 4. Discussion

In our retrospective study we analyzed the thyroid disorder spectrum in the children aged from birth to 16 years following in pediatric endocrinology clinic in the teaching hospital over a 7-year period. A total of 148 patients most of our patients were Saudis (84.9%), majority of them were females at 64.2%. In the literature, almost all the different thyroid disorders are affecting female more commonly than males at all ages [3], in comparison to other studies the overall female ratio is higher in our study. 

The most prevalent thyroid disease in children is Hashimoto’s related hypothyroidism (50.34%), which matches the international studies [2,3,5], acquired hypothyroidism; both Hashimoto’s and non-Hashimoto’s nearly accounted twice the congenital hypothyroidism unlike other studies, which suggests that congenital hypothyroidism is more prevalent than acquired forms of hypothyroidism [15]. Acquired hypothyroidism in form of autoimmune thyroiditis, is the most common cause of hypothyroidism in children in the United States. The treatment requires oral levothyroxine aiming for the reduction or absence of the clinical symptoms and signs of hypothyroidism, retain normal physical parameters, such as weight and linear growth, and to normalize TSH levels to the age-appropriate range [16]. Levothyroxine treatment might be continued for life; however, DeLuca et al. found that children with acquired hypothyroidism became euthyroid in about 50% of cases, suggesting that levothyroxine may eventually be discontinued [17]. In cases of persistent goiter, increasing levels of anti-thyroid peroxidase antibody or serum TSH mandates the continuation of the thyroxine replacement therapy.

Hashimoto’s thyroiditis was the most common cause of hypothyroidism [18], followed by congenital hypothyroidism (26.4%), Hashimoto’s thyroiditis (17%), subclinical hypothyroidism (10.4%), hyperthyroidism (3.8%), thyroglossal duct cyst (3.8%), thyroid nodule (3.8%), and subclinical hyperthyroidism (0.9%). These patients were categorized according to the age at diagnosis and in relation to the disease into three age groups, first from birth–6 years old (46.2%), the second age group from 7 years old to 12 years old (39.6%). The third age group from 12 years old to16 years old (14.2%). Of the patients who were diagnosed from birth to 6 years old, most of them were having congenital hypothyroidism (57.1%), mainly as primary congenital hypothyroidism in 94%, which is similar to a local study in the western region of Saudi Arabia [19].The next common was acquired non-Hashimoto’s hypothyroidism (24.5%), followed by Hashimoto’s thyroiditis and subclinical hypothyroidism both were equal in this age group (8.2%), and lastly, one patient was diagnosed with thyroglossal duct cyst in this age group that contribute to 2%, these percentages are found to be like the similar published studies in different countries [20].

The second age group who were from the age of 7 years old to 12 years old (39.6%), most of them were diagnosed with acquired hypothyroidism (45.2%), including Hashimoto’s thyroiditis (23.8%), hyperthyroidism and subclinical hypothyroidism were equal in this age group (9.5%), thyroid nodule (7.1%), and finally, subclinical hyperthyroidism and thyroglossal duct cyst were equal in percentage (2.4%).

In contrast, thanks to the newborn screening program, all the patients of congenital hypothyroidism were diagnosed at the first age group from birth to 6 years of age (100%), notably, this is highly important for preventing complications specifically the neurocognitive function and intellectual development as shown in some literature that emphasis in the relation between congenital hypothyroidism and intellectual development [21]. Our hospital unpublished incidence of congenital hypothyroidism is approximately 1:2400, with female predominance, which is comparable to local study findings of 1:2470, generally it is higher than the average international incidence 1:300–4000, possibly related to high incidence of consanguinity marriage in Saudi Arabia [22].

Most of the patients with Hashimoto’s thyroiditis were diagnosed at the second age group from 7 years old to 12 years old (55.6%), at this second age group we got all the patients with non-Hashimoto’s acquired hyperthyroidism. Majority of the patients with acquired hypothyroidism were diagnosed at the second age group from 7 years old to 12 years of age, which is comparable to Edo Hasanbegovic et al. study [23], we had one patient with hyperthyroidism at this second age group. 

The first age group from birth to 6 years of age and the second age group from 7 years old to 12 years of age were found to have the same number of patients diagnosed with subclinical hypothyroidism at these two age groups. Although, for children, subclinical hypothyroidism is often a benign and remitting condition with no long-term serious effect of high TSH level in these subjects [24].

Most of the patients with the patients with thyroglossal duct cyst TGDC were diagnosed at the third age group from 12 years old to 16 years of age (50%), TGDC considered the commonest cause of congenital neck mass in children [25], in literature; most of the cases are diagnosed before age of 30 years, with more than half of them are detectable clinically before age of 10 years, in comparison; our cases with TGDCs were diagnosed relatively at older age [26].

Most of the patients with thyroid nodule were diagnosed at the second age group from 7 years to 12 years of age (75%), although we do not have their surgical follow up results, generally, most of thyroid nodules in pediatric age group are benign; however, it has a 2–3-fold increased risk of malignancy compared to adults. Considering the small size of our sample, this age group is relatively comparable to the published studies. However, we had more males in our study 4:1, which is higher than the reported ratios of female to males 1.5:1 below the age of 15 years, and increase to 3:1 in the age of 15–20 years. [27]. 

Presence of antibodies is characterized by having either positive antithyroglobulin antibody test (normal range ≤ 4.11 IU/mL), or positive thyroid peroxidase antibodies test (normal range ≤ 5.61 IU/mL), or both. Regarding the presence of antibodies, a total of 54 patients were found to have positive antibodies which represent 36.8% of total patients, this percentage is close to some studies [9,10,11,12], however they are higher than other similar studies [12,13,14]. The most common disorder associated with the presence of antibodies was Hashimoto’s thyroiditis [28].

Correlating Hashimoto’s thyroiditis to each specific antibody, we found 16 out of 18 patients diagnosed with Hashimoto’s thyroiditis 88.9% were found to have positive antithyroglobulin antibody test. A total of 13 out of 18 patients diagnosed with Hashimoto’s thyroiditis (72.2%) were found to have positive thyroid peroxidase antibody test, which has been noticed that there is a rising incidence of autoimmunity in young people [6].

Regarding thyroid disorder association with other morbidities, we got almost half of the affected patients had other comorbidities. The most frequently encountered is Down Syndrome followed by bronchial asthma. This is inconsistent with some of the literature which suggest that Hashimoto’s Thyroiditis is the most frequently encountered in patients with Down Syndrome, while in our study there were no case of Hashimoto’s Thyroiditis that is associated with Down Syndrome, and the literature emphasis in the annual screening of thyroid spectrum disease in those susceptible patients [29].

We found that type 1 Diabetes Mellitus T1DM as the major thyroid disorder comorbidity together with bronchial asthma both constitutes 60% of all comorbidities. In one study, they found half of the newly diagnosed T1DM had abnormal thyroid function tests, which was normalized after insulin initiation therapy; however, the positive antithyroid antibodies were more predictive of abnormal thyroid in T1 diabetic patients [30], and they recommend to check for thyroid antibodies at diagnosis of T1DM rather than thyroid function tests.

Although autoimmune disorders and allergic diseases are caused by an impaired immunological reaction by different types of T-helper cells, the literature did not show strong association or increase in the risk of developing each other. Furthermore, Dilber et al., found the presence of Hashimoto’s thyroiditis might be protective and ameliorative of symptoms in asthmatic patients. [31], our findings of high asthmatics might be coincidental since this genetic disease is common in our high consanguinity rate area and has multiple environmental factors.

Growth-related was the third most frequently encountered (18.9%) in the literature. Growth complaints of short stature and weight gain are of major issue especially in acquired Hypothyroidism. This also found in our data that agrees with international studies in tackling abnormalities that interfere with growth velocity and the final adult height of the affected children [32].

We had one patient having autism associated with thyroiditis, psychological, and developmental comorbidities, such as autism and depression might be associated with a spectrum of thyroid abnormalities, including clinical and subclinical thyroiditis; however, there is no cause-and-effect relationship. In one study by Cythia et al., regression in autistic children was significantly associated with a family history of autoimmune disorders [33].

### Limitations and Recommendations

We acknowledged the retrospective study nature of our study. Furthermore, the sample size is relatively small. We encountered some of the missing data that might be of help in finding other cofactors and manifestations of the specific thyroid disease in each age group.

## 5. Conclusions

This study showed the spectrum of variant thyroid disorders in the pediatric endocrine clinic at a teaching hospital in Saudi Arabia. Thyroid disorders spectrum in our study is highly comparable to the local and international findings, apart from the inherited disorders, which are common in our community with high consanguinity marriages. Thyroid disorders affect all the pediatric age groups, having different impact on each parameter of their intellectual, physical, and behavioral development. This mandates the need for early diagnosis and treatment with regular follow up to prevent complications of these disorders. Screening programs in the newborn for congenital hypothyroidisms is highly recommended to detect these disorders and treat them earlier to prevent the devastating intellectual disability. 

Screening of the autoimmune thyroid disorders among the pediatric patients have other autoimmune diseases is effective clinical strategy when they are caught in the early phase of the thyroid disease and prevents progression and complications. Further international studies of thyroid disorder in pediatric age group might give different perspectives regarding the pattern and variation of thyroid disease spectrum.

## Figures and Tables

**Figure 1 children-10-00390-f001:**
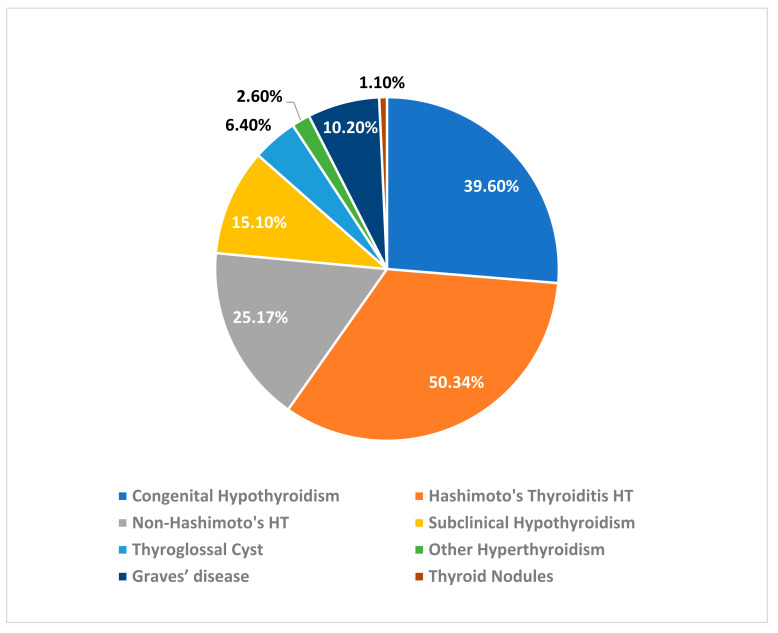
The percentage of each thyroid disorder among all affected pediatric patients. HT: hypothyroidism.

**Figure 2 children-10-00390-f002:**
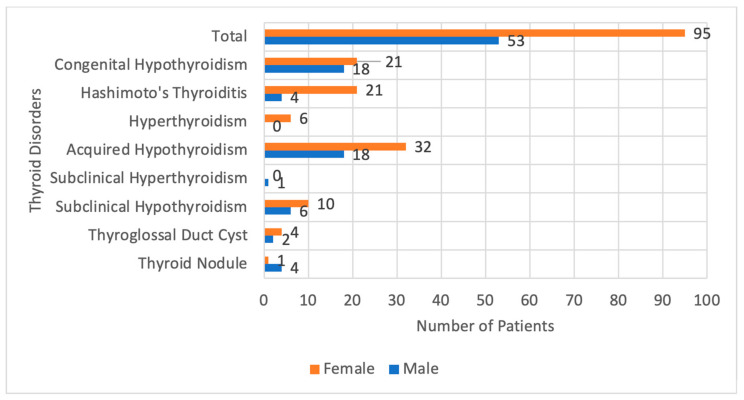
Gender Distribution of Each Specific Thyroid Disorders; showing female predominance in all thyroid disorders except thyroid nodules being commoner in males.

**Table 1 children-10-00390-t001:** Gender distribution of the affected patients.

Gender	Frequency	Percentage
Males	53	35.8%
Females	95	64.2%
Total	148	100.0%

**Table 2 children-10-00390-t002:** Age group and gender distribution of thyroid disorders.

	Birth–6 Years (46.20%)	7–12 Years (39.6%)				13–16 Years (14.6%)
	Percentage of the Group	Females (%)	Percentage of the Group	Females (%)	Percentage of the Group	Females (%)
Congenital Hypothyroidism	57.10%	66	None	None	None	None
Hashimoto’s Thyroiditis	24.5%	60	47.70%	65	33.30%	74%
Non-Hashimoto’s Thyroiditis	8.2%	72	23.80%	58	26.70%	68%
Subclinical hypothyroidism	8.2%	78	9.50%	52	20%	59%
Graves’ Disease	None	None	9.50%	88	13.30%	89%
Thyroglossal Duct Cyst	2%	100	2.40%	100	None	None
Thyroid Nodules	None	None	7.10%	25	6.70%	40%

Note. The percentage of females’ distribution is shown in brackets (2.40%).

**Table 3 children-10-00390-t003:** Referral Causes by system.

Category	Number of Cases
Screening Tests (28.4%)	
Congenital hypothyroidism	30
Autoimmune disorders	12
Localized Symptoms/Signs (22.4%)	
Neck Mass	29
Respiratory symptoms	4
Physical Growth (18.9%)	
Weight gain	16
Short stature	10
Weight loss	2
Metabolic/Energy (16.9%)	
Fatigue	7
Poor feeding/low appetite	6
Sweating	3
Palpitation	3
Cold intolerance	2
Heat intolerance	2
Chronic constipation	2
Dermatological (7.4%)	
Hair loss	8
Dry skin	3
Incidental (4%)	
Abnormal laboratory tests	6
Female puberty (2%)	
Irregular periods	3

Note. Each system expressed in percentage out of the total patients’ sample in brackets.

**Table 4 children-10-00390-t004:** Comorbidities associated with acquired hypothyroidism.

Category	Comorbidities	Number of Cases
Syndromic	Down Syndrome	8
Rheumatological Disorders	Systemic Lupus Erythematosus	1
Congenital anomalies	Cleft Palate	1
Psychiatric	Autistic Spectrum	1
Metabolic	Dyslipidemia	1
Autoimmune disorders	Type 1 Diabetes Mellitus	7
Others	Iron Deficiency Anemia	1

## Data Availability

Data will be available on request from the corresponding author.

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
