# Peer review of "Thyroid Disorders Spectrum in Pediatric Endocrine Clinic; Seven-Year Experience of a Teaching Hospital in Saudi Arabia"

_children, 2023, doi:10.3390/children10020390_

Round 1

Reviewer 1 Report

The authors propose to determine the prevalence of thyroid disorders  among pediatric patients visiting the endocrine clinic in the teaching hospital; differentiate the spectrum of different thyroid disorders in gender perspectives and age groups of the children and evaluate the effect of associated disorders and the  major risk factors of thyroid disorder in pediatric age group.

This is a long retrospective study (2015-2021) where authors analyzed data from 148 patients and described the different thyroid disorders in children that were diagnosed at their center in this period. Nothing original is brought from this manuscript, not addressing any gap in the field. All the diseases and prevalence match with what is known in thyroid disorders in children.

The main conclusion refers to the local population. Since there this is a no-novel study, I recommend that authors perform a systematic analysis and incorporate their cohort in it. Otherwise, this manuscript is for locals only.

Figures 1 and 2 are inverted. Figure 2 is the first, so it has a broad amount of information. Congenital hypothyroidism is of utmost importance and should appear as the main result. Moreover, the occurrence of chromosomal disorders should be discussed considering international literature and local culture.

Table 2 is unnecessary. Should be suppressed.

If you only have one case this is a case discussion, and you cannot perform any type of statistic with it.

A broad English review should be performed by specialized and native English speakers.

The results and discussion are no news and should be incorporated in a broad collection of data – a systematic review.

Author Response

Thank you so much for your effort and kind review of our manuscript.

We appreciated and welcomed all your comments and suggestions and followed them in our revised manuscript.

Please see the attached document for the point by point response to your valid comments and queries.

Best regards

1- This is a long retrospective study (2015-2021) where authors analyzed data from 148 patients and described the different thyroid disorders in children that were diagnosed at their center in this period. Nothing original is brought from this manuscript, not addressing any gap in the field. All the diseases and prevalence match with what is known in thyroid disorders in children.

We have added (in Saudi Arabia) in the title to give an idea of the local nature of our findings, we also compared it to local and international published findings

2- The main conclusion refers to the local population. Since there this is a no-novel study, I recommend that authors perform a systematic analysis and incorporate their cohort in it. Otherwise, this manuscript is for locals only.

Yes, it is local, however there are very few published studies regarding the outpatient (OPD) spectrum of thyroid disorders to give an idea about which disorders are common and which are less and how this can improve their outcome and plan the future resources.

3- Figures 1 and 2 are inverted. Figure 2 is the first, so it has a broad amount of information.

Figures have been switched to the proper sequence, in addition corrected in the text in red

4- Congenital hypothyroidism is of utmost importance and should appear as the main result. Yes, we have added more in this regard both in results and discussion as well as in the introduction.

5- Moreover, the occurrence of chromosomal disorders should be discussed considering international literature and local culture.

We already added more details regarding Down syndrome since it is the commonest chromosomal disorder associated with thyroid pathologies.

6- Table 2 is unnecessary. Should be suppressed.

   We reconstructed and shrunken Table 2

7- A broad English review should be performed by specialized and native English speakers.

    Rephrased and rewritten.

8- The results and discussion are no news and should be incorporated in a broad collection of data – a systematic review.

 We added more discussion regarding our results and compare them to local and international findings.

Reviewer 2 Report

Al-Qahtani and colleagues reviewed medical records and catalog the various thyroid disorders that they encountered in children from newborns through age 16 who were referred to their clinic in a teaching hospital in Saudi Arabia from January 2015 through December 2021.

Comments

11.     Line 103. It is unclear what Endocrine tests were performed to substantiate the various diagnoses. Each diagnosis should be defined. Lines 254 ff and 260-263 should be presented in Results.

22      Line 118 and 280.  Hashimoto thyroiditis is an acquired form of hypothyroidism (line 65).  By “acquired hypothyroidism” do the authors refer to patients with negative results for anti-microsomal and anti-thyroglobulin antibodies? The Discussion on lines 258 implies that some patients with “acquired hypothyroidism” were antibody positive.  Clarification is needed.

33.    Line 73.  The newborn screening program should be described.

44.       Were patients with iodine deficiency encountered as in the Anseer region of Saudi Arabia?  (Abbag F, Int J Environ Res Pub Health, July 2021).

55.       Lines 122, 129, 212, 218, 230.  Subclinical (mild) hypo- or hyper-thyroidism are not clinical entities but rather define the severity of thyroid disorders based on mild laboratory abnormalities some of which might spontaneously remit while others progress or remain unchanged.  Children presenting with mild (subclinical) thyroid disorders should be discussed separately.

66.       Six patients were diagnosed with a thyroid nodule. How were they evaluated? What was the final diagnosis?

77.       Line 43. “Isolated ….developed countries”.  A reference is needed. Are there published data for Saudi Arabia or other countries in the middle- East?

88.       Line 115. “patients were following regularly” Were all patients seen at the Clinic included in the analysis, or only those “followed regularly” which should be defined.

99.       Line 123. More details of the patients with Hashimoto thyroiditis could be added;  e.g. goiter prevalence, euthyroid vs hypothyroid, history of hyperthyroidism.

110.   Figure 1 is not of publication quality- it should be redrawn.

111.   Figure 1. The list of diagnoses on the left and the number of columns of results are inconsistent. The legend should explain the data in the figure.

112.   Figure 1.  Subclinical hypothyroidism should be removed, and congenital hypothyroidism should be added to this figure. The differences by sex should be analyzed statistically, and noted as symbols in the figure.

113.   Lines 130-1 are confusing, and should be rewritten.

114.   Line 136 ff. It is difficult to follow the distribution of diagnoses by age.  A Figure showing the 3  age  groups along the top  and the possible diagnoses vertically with the number of subjects (M/F)  affected in each box might more clearly convey the authors’ findings.

115.   Line 170. The presentation of symptoms might be reorganized to follow the order in the table.

116.   Much of the Discussion repeats the Results section, and should be revised and reorganized.  Instead, the significance of the data should be discussed in the context of what has been published by others.  e.g. The comparison with surveys from other races and ethnicities might be expanded. What is the prevalence of congenital hypothyroidism in this population versus others (Abbas M et al. Cureus. 12:e7166, 2020). The connection between asthma and thyroid disease (line 267) is not well known and should be discussed.  Age of presentation of Hashimoto thyroiditis may differ from earlier studies (ref 3; Erbas IC et al, J Ped Endocrinol Metab 2021).

117.   Lines 204-5. Reference 3 is a review of Hashimoto thyroiditis in children. It does not seem to apply to this sentence.

118.   Lines 209-11 are confusing.   Hashimoto thyroiditis is mentioned twice.

119.   There are many English language and usage errors. Professional editing is needed.

220.    Many references are missing.

Minor Comments

21.   Line 38 and throughout.” …in pediatric age group, thyroid disorder is…” Each sentence should end with a period and be followed by 2 spaces.

22.   Figures. There are 2 titles. One in the legend, and second in the figure.

23.   Figure 1. Relation.

24.   Lines 140 and 149 should begin new paragraphs

25.   Line 156. acquired hyperthyroidism. ?

26.   Line 167, the subspecialty of an offspring ?

27.   Table 3. The reason for the two-column format is unclear.

28.   The format in the reference list is inconsistent

29.   Reference 3 is incomplete. Superscripts are not needed.

310.   Reference 7. The authors were omitted.

Author Response

Thank you so much for your effort and kind review of our manuscript.

W appreciated and welcomed all your comments and suggestions and followed them in our revised manuscript.

Please see the attached document for the point by point response to your valid comments and queries.

Best regards

Comments and Suggestions for Authors

Thank you so much for your valuable suggestions and appropriate comments, and I am happy to get my response to them as follows

  1.   Line 103. It is unclear what Endocrine tests were performed to substantiate the various diagnoses. Each diagnosis should be defined. Lines 254 ff and 260-263 should be presented in Results.

      For line 103; definitions of all thyroid disorders have been added to the revised manuscript text in green.

     For line 254, 260-263, we presented them to the results section in green and kept the related comparison in the discussion section as well.

2      Line 118 and 280.  Hashimoto thyroiditis is an acquired form of hypothyroidism (line 65).  By “acquired hypothyroidism” do the authors refer to patients with negative results for anti-microsomal and anti-thyroglobulin antibodies? The Discussion on lines 258 implies that some patients with “acquired hypothyroidism” were antibody positive.  Clarification is needed.

      Very valid point, we have added the subtypes of all the acquired hypothyroidism as  Hashimoto’s, and non-Hashimotos, and based on that we have rephrased the results and the discussion, furthermore we recreated the pie-chart graph showing the different subtypes in new percentage after subtyping     ( Figure 1).

       We crossed the statement in line 258 in red

  1.   Line 73. The newborn screening program should be described.

The screening program method has been added to the text in green

  1.      Were patients with iodine deficiency encountered as in the Anseer region of Saudi Arabia?  (Abbag F, Int J Environ Res Pub Health, July 2021).

      Aseer area as most of the Southwest of Saudi Arabia are mountainous regions and prone to have Iodine and iodine related disorders, which we don’t use to have in the Eastern area in Saudi Arabia. However, this hasn’t been studied well. In our study we did not have this diagnosis in our patients’ records.

  1. Lines 122, 129, 212, 218, 230.  Subclinical (mild) hypo- or hyper-thyroidism are not clinical entities but rather define the severity of thyroid disorders based on mild laboratory abnormalities some of which might spontaneously remit while others progress or remain unchanged.  Children presenting with mild (subclinical) thyroid disorders should be discussed separately.

     We totally agree with you from the clinical perspective point of view; however, we got all these diagnosis as they are coded under the thyroid disorders in the hospital record system.

  1. Six patients were diagnosed with a thyroid nodule. How were they evaluated? What was the final diagnosis?

     Any patient with confirmed thyroid nodules is referred to our endocrine surgery clinic, were they got the confirmatory FNAs and management accordingly. Our surgeon colleagues are planning to submit a separate paper for that purpose.

  1. Line 43. “Isolated ….developed countries”.  A reference is needed. Are there published data for Saudi Arabia or other countries in the middle- East?

  Reference has been added. In general Iodine-sufficient regions in the world ( developed countries) have the autoimmune thyroid disorders more common than iodine related thyroid disorders, which are common in the underdeveloped countries. No specific epidemiological study in the Saudi population

  1. Line 115. “patients were following regularly” Were all patients seen at the Clinic included in the analysis, or only those “followed regularly” which should be defined.

    All the clinic logbook patients were included, however as a finding in their medical records they were found to be generally regularly followed in the clinic (defined by every 3 months appointment). Important point: it was not decided as inclusion or exclusion criterion.

  1. Line 123. More details of the patients with Hashimoto thyroiditis could be added;  e.g. goiter prevalence, euthyroid vs hypothyroid, history of hyperthyroidism.

      Details have been added in red.

  1. Figure 1 is not of publication quality- it should be redrawn.

     Format and size of of the Figure1 have been adjusted to make the graphic data much clear

  1. Figure 1. The list of diagnoses on the left and the number of columns of results are inconsistent. The legend should explain the data in the figure.

     Legend has been added explaining the major findings

  1. Figure 1.  Subclinical hypothyroidism should be removed, and congenital hypothyroidism should be added to this figure. The differences by sex should be analyzed statistically, and noted as symbols in the figure.

      Congenital hypothyroidism is already added in the graph, legend explains gender differences in the affected thyroid disorders

  1. Lines 130-1 are confusing, and should be rewritten.

       Statement has been rephrased

  1. Line 136 ff. It is difficult to follow the distribution of diagnoses by age.  A Figure showing the 3  age  groups along the top  and the possible diagnoses vertically with the number of subjects (M/F)  affected in each box might more clearly convey the authors’ findings.

Thanks for this advice, we did it in a new comprehensive table number (4)

  1. Line 170. The presentation of symptoms might be reorganized to follow the order in the table.

      The presentations in the text have been rearranged (in red) according to their percentage and sequence in the table 3

  1. Much of the Discussion repeats the Results section, and should be revised and reorganized.  Instead, the significance of the data should be discussed in the context of what has been published by others.  e.g. The comparison with surveys from other races and ethnicities might be expanded. What is the prevalence of congenital hypothyroidism in this population versus others ( ). The connection between asthma and thyroid disease (line 267) is not well known and should be discussed.  Age of presentation of Hashimoto thyroiditis may differ from earlier studies (ref 3; Erbas IC et al, J Ped Endocrinol Metab 2021).

    We already explained more in the methodology and added more in discussion, with comparison to recent local and international results, we also added new recent local references as well.

  1. Lines 204-5. Reference 3 is a review of Hashimoto thyroiditis in children. It does not seem to apply to this sentence.

     This has been corrected and new reference has been replaced in red.

  1. Lines 209-11 are confusing.   Hashimoto thyroiditis is mentioned twice. (to revise all results and redraw the category pie chart)
  2. There are many English language and usage errors. Professional editing is needed.

      Done

  1. Many references are missing.

      We added some extra references

Minor Comments

  1. Line 38 and throughout.” …in pediatric age group, thyroid disorder is…” Each sentence should end with a period and be followed by 2 spaces.

     Done in red

  1. Figures. There are 2 titles. One in the legend, and second in the figure.

      Have been revised and corrected

  1. Figure 1. Relation.

     Has been corrected in the graph

  1. Lines 140 and 149 should begin new paragraphs

     Done in red

  1. Line 156. acquired hyperthyroidism. ?

     Has been explained in red

  1. Line 167, the subspecialty of an offspring ?

    Corrected and explained in detail, in red

  1. Table 3. The reason for the two-column format is unclear.

  Reconstructed in one-column format

  1. The format in the reference list is inconsistent
  2. Reference 3 is incomplete. Superscripts are not needed.

     Rewritten in red

  1. Reference 7. The authors were omitted.

All authors named have been retained in red in the references

Round 2

Reviewer 1 Report

I thank the authors for the changes that improved the manuscript. 

Author Response

Thank you for your valid comments and suggestions.
We did our best in following all the suggested points; including the professional tables and figures reconstructions, English editing and the specific suggested changes in specific sections in the manuscript.